# Fish Oil Supplementation with Resistance Exercise Training Enhances Physical Function and Cardiometabolic Health in Postmenopausal Women

**DOI:** 10.3390/nu15214516

**Published:** 2023-10-25

**Authors:** Sang-Rok Lee, Dean Directo

**Affiliations:** Department of Kinesiology, New Mexico State University, Las Cruces, NM 88003, USA; ddirecto@nmsu.edu

**Keywords:** resistance exercise training, fish oil, physical function, blood pressure, inflammation, oxidative stress

## Abstract

Menopause is a condition associated with an increased risk of dysregulation in cardiovascular and metabolic health among older women. While fish oil (FO) has garnered great attention for its health-enhancing properties, its potential for enhancing cardiometabolic health in this demographic remains to be established. The purpose of this study was to determine the clinical efficacy of an 8 wk administration of FO combined with programmed resistance exercise training (RET) on physical function and risk factors associated with cardiometabolic health in healthy older women. Twenty, healthy, older women were randomly assigned to one of the two experimental groups: resistance training with placebo (RET-PL) or RET with fish oil (RET-FO). Physical function, blood pressure (BP), triglyceride (TG), and systemic inflammation and oxidative stress biomarkers were assessed before and after the intervention. Statistical significance was set at *p* ≤ 0.05. Physical function was greatly enhanced in both RET and RET-FO. Handgrip strength substantially increased only in RET-FO. RET-FO exhibited significant decreases in BP, TG, inflammatory cytokines (TNF-α and IL-6), and oxidative stress (MDA and 8-OHdG) levels, while no detectable changes were found in RET-PL. Our findings indicate that FO administration during 8 wks of RET appears to enhance muscle function and lower risk factors linked to cardiometabolic disorders in postmenopausal women.

## 1. Introduction

With age, humans inevitably experience changes in cardiometabolic health and a decline in muscle mass (i.e., sarcopenia), which may contribute to an increased incidence of physical disability [1,2] and a loss of independence in activities of daily living (ADL), leading to health disparities among older adults. In premenopausal women, abundant levels of primary female sex hormones, such as estrogen, provide cardiometabolic protective properties. However, there is a gradual decline in this hormone with age, potentially rendering older women at higher risk for cardiometabolic disorders. This decline in estrogen may lead to the dysregulation of lipid metabolism, cardiovascular diseases, metabolic disorders, and obesity [3]. Research demonstrates a negative correlation between estrogen levels and triglyceride (TG) concentrations, with a notable increase in TG levels during menopause [4].

Key features of aging encompass elevated levels of reactive oxygen species (ROS) and chronic low-grade inflammation, which are potential pathophysiologic factors associated with cardiometabolic disorders and the accelerated sarcopenic process [5,6,7]. Heightened oxidative stress can potentially advance cardiometabolic disorders, including various types of cardiovascular diseases, hypertension, and metabolic syndromes (e.g., type 2 diabetes) [8], as well as diverse diseases linked to muscle wasting [9]. The up-regulation of inflammatory cytokines such as tumor necrosis factor-alpha (TNF-α) and interleukin-6 (IL-6) also contribute to the development of cardiometabolic disease [10,11] and muscle loss [12,13,14].

Resistance exercise training (RET) is acknowledged as the most cost-effective counteractive strategy for enhancing muscle mass, improving physical function, and reducing morbidity in older adults [15]. RET retards the sarcopenic process by stimulating the activation of the skeletal muscle anabolic mediators [i.e., Phosphatidylinositol-3-kinase (PI3K)/Akt and mammalian target of rapamycin (mTOR) cascade] [16], while concurrently down-regulating primary mediators of proteasomal degradation [17]. Evidence suggests that chronic RET reduces inflammatory mediators like TNF-α [18], while enhancing protein synthesis in the skeletal muscle of older adults [19]. However, there is conflicting evidence regarding the association between RET and blood lipid profiles. A study by Tucker and Silvester found a reduced risk of dyslipidemia among individuals who participated in RET programs [20]. On the contrary, other studies have indicated that RET intervention does not significantly affect lipid profiles in middle-aged or older adults [21,22]. In addition, Kohl and colleagues reported no significant correlation between muscle strength and cholesterol levels in both males and females [23].

Omega-3 polyunsaturated fatty acids (*n*-3 PUFA), composed of eicosapentaenoic acid (EPA) and docosahexaenoic acid (DHA), possess lipolytic [24], anti-inflammatory [25], and antioxidant [26] properties, attenuating the activity of inflammation and oxidative stress mediators secreted from adipose tissue. Chronic *n*-3 PUFA supplementation has been shown to enhance muscle mass and strength in older adults [27]. Furthermore, *n*-3 PUFA administration has been associated with a reduction in inflammation. Given the implication of chronic inflammation in the sarcopenic process and cardiometabolic disorders with aging, *n*-3 PUFA may serve as a potential therapeutic candidate for older women. The primary aim of the present study was to determine the efficacy of *n*-3 PUFA administration during 8 wks of programmed RET on physical function, blood pressure (BP), fat metabolism, and systemic inflammatory and oxidative stress biomarkers in postmenopausal women. We hypothesized that *n*-3 PUFA administration would enhance the benefits derived from 8 wks of programmed RET in older females.

## 2. Materials and Methods

### 2.1. Participants and Experimental Design

This study was approved by the New Mexico State University Institutional Review Board (no. 17616) on January 15, 2019. Twenty healthy older women (65.65 ± 3.39 y) were recruited for this study. Group sizes were determined by a-priori power analysis using effect size calculated from previous evidence which showed notable differences in muscle strength following exercise in older adults [28]. Inclusion criteria comprised women who: (1) had reached menopause; (2) were healthy, without any physical or mental disorders that could contraindicate completion of the proposed exercise intervention; (3) were a non-smoker; (4) had not taken *n*-3 PUFA or antioxidant supplements in the 12 months preceding the present study; (5) were not using anti-inflammatory drugs; (6) were not engaged in resistance exercise training in the 12 months that preceded the present study; and (7) did not consume excessive alcohol (no more than five drinks per week). All participants underwent thorough screening before enrollment, and, upon meeting the criteria, provided written informed consent prior to participating in the study.

A longitudinal design was employed to examine changes in all dependent variables from pre- to post-intervention. Participants visited the laboratory for pre-intervention assessments, including evaluations of grip strength, physical function, BP, TG, and blood biomarkers for systemic inflammation and oxidative stress. After completing all baseline assessments, participants were randomly assigned to either resistance exercise training and a placebo supplement (RET-PL, n = 10) or RET combined with FO supplementation (RET-FO, n = 10) for an 8 wk intervention. Upon completion of the 8 wk intervention, they returned to the lab for post-intervention outcome assessments. Throughout the 8 wk intervention period, all participants were instructed to maintain their regular diet, normal daily activity, and regular sleep schedule. They were advised to refrain from any strenuous exercise or physical activity (e.g., resistance exercise training held outside of the research laboratory) and to avoid consuming food (e.g., high-fat diet) or beverages (e.g., coffee, tea, etc.) that could affect their metabolic rate for 48 h before their lab visit to minimize the variability of the dependent variables.

### 2.2. Resistance Exercise Training

All participants engaged in programed RET twice per week for 8 wks. The exercise regimen involved 3 sets of 12 repetitions or until failure (whichever came first) for 5 major muscle groups in the upper and lower body (lat pull-down, seated row, biceps curl, leg press, calf raise). The training sessions were closely supervised by a member of the research team to ensure proper execution and minimize the risk of injury during the RET sessions. The exercise intensity was initially set at 50% of their one-repetition maximum (1 RM). Subsequently, the training load was increased to 70% of 1 RM in the second week and gradually raised (by 5% weekly if participants successfully completed the given workload) to promote an adaptive hypertrophic response. If a participant could not complete the assigned workload of 10 RM, the same workload was maintained for the following session. Each exercise session commenced with a warm-up involving stretching and one set of low-intensity exercise (30% of 1 RM, 10 RM). The details of each training session, including load increments, were meticulously recorded in a logbook.

### 2.3. Fish Oil Supplement

The FO supplement comprised a blend of eicosapentaenoic acid (EPA; 0.7 g) and docosahexaenoic acid (DHA; 0.24 g). The supplement group ingested 3 FO capsules, one with each meal, providing a total of 2.1 g EPA and 0.72 g DHA per day. For the RET-PL group, identical-looking placebo capsules containing safflower oil were administered at a dose of 3 capsules per day. The dose for the FO supplement has received approval from the Food and Drug Administration (FDA) for effectively reducing TG concentrations in individuals with high TG levels [29]. Moreover, a prior study reported a significant increase in whole-body protein synthesis with similar doses in rodent animals (relative to their body weight) [30]. Participants were instructed to consume the supplement pills daily and return any empty or remaining pills to monitor their compliance. Tablet counts revealed compliance rates exceeding 95% in each group.

### 2.4. Physical Function Assessment

Handgrip strength measurements were conducted both before and after the intervention to assess changes in muscle strength with each group. An analog hand dynamometer was utilized for the handgrip strength assessment. Participants stood with their elbow at a right angle by their side, maintaining a neutral wrist position. The base grip was positioned on the first metacarpal, with four fingers placed on the handles. Upon readiness, participants exerted their maximal effort to squeeze the dynamometer for 5 s without moving their body. Three trials were performed, and the highest recorded number was noted.

Physical function related to leg strength and activities of daily living was evaluated using the following tests: (1) Five times sit-to-stand (5X-STS): participants executed this test using a hard, straight-backed, arm-less chair (43 cm in height) placed against a wall. With both arms crossed on the chest, they completed three trials of five sit-to-stand repetitions at their maximum speed with a 2 min rest interval. The time was recorded upon the completion of the fifth repetition. (2) Timed up and go (TUG): participants were instructed to sit on a chair (43 cm height), rise, walk around a cone (2.44 m from the chair), and return back to the chair at their maximum walking speed. Three trials were performed with a 1 min resting interval. (3) 6 m walk (6MW): participants walked to a marked point over 6 m at their maximum walking speed for three trials with a 30 s rest between each trial. (4) 30 s sit-to-stand (30S-STS): participants were tasked with repeatedly standing up and sitting down from a chair as many times as possible within 30 s. The number of repetitions was recorded.

### 2.5. Blood Pressure Assessment

Resting arterial BP was measured both before and after the intervention using an electronic blood pressure sphygmomanometer (Panasonic EW31092, Newark, NJ, USA). Each participant sat in a quiet environment for 10 min, and the BP of their left arm, positioned on a table at heart level, was measured twice. Mean values for systolic blood pressure (SBP) and diastolic blood pressure (DBP) were recorded. If the two measurements provided values that differed by more than 2%, a third measurement was taken, and the mean value of the two closest measurements was selected for the data analysis. Mean arterial pressure (MAP) was calculated using the following equation, MAP = [(2/3 × DBP) + (1/3 × SBP)].

### 2.6. Blood Collection and Analysis of Systemic Biomarkers

Participants were instructed to fast at least eight hours prior to their scheduled blood draw for the biochemical assessments. Blood samples (10 mL) were collected from each participant’s forearm vein using a 23-gauge needle and vacutainer tubes containing EDTA pre- and post-interventions. The collected samples were promptly centrifuged to separate plasma and stored in multiple aliquots at −80 °C for the analysis of relative biomarkers. Triglyceride (TG) levels were determined using a triglyceride colorimetric assay kit (Cayman Chemical Company, Ann Arbor, MI, USA). IL-6 and TNF-α were assessed using commercially available ELISA kits (Raybiotek, Atlanta, GA, USA). Malondialdehyde (MDA) levels were assessed using a colorimetric assay kit (Northwest Life Science Specialties, Vancouver, WA, USA). 8-hydroxy-2′-deoxyguanosine (8-OHdG) was analyzed using a DNA/RNA oxidative damage ELISA kit (Cayman Chemical Company, Ann Arbor, MI, USA). All samples were analyzed in duplicate.

### 2.7. Statistical Analysis

Statistical analysis was conducted using SPSS software (SPSS version 25, IBM). All data are presented as the mean ± standard deviations (SD). Following confirmation of normality, 2 (experimental condition) × 2 (time) repeated measure ANOVA was employed. Cohen’s *d* effect size was evaluated using the equation: d = (mean difference between pre- and post-intervention)/(pooled standard deviation). According to Cohen’s conventions, effect sizes of 0.2, 0.5, and 0.8 were considered small, medium, and large effects, respectively. Cohen’s *d* values ≥ 0.5 were interpreted as a practical or functional impact of intervention.

## 3. Results

Descriptive data for research participants are provided in Table 1. Pre-intervention analysis indicated no notable differences in age and anthropometric characteristics between the groups. All participants successfully completed the given intervention, showing a consistent progression of exercise intensity over the 8 wk experimental period. The workload remained consistent and was not significantly different between the experimental groups throughout the entire intervention.

### 3.1. Physical Function and Blood Pressure

The data for physical function and BP are presented in Table 2. There were significant time effects for all aspects of physical function. Initially, there was no difference in handgrip strength between the experimental groups at baseline. However, handgrip strength significantly increased in REF-FO from the baseline (+5.7%, *p* < 0.001, *d* = 0.30), while RET-PL exhibited a tendency towards increased grip strength from the baseline (+2%, *p* = 0.060, *d* = 0.10). In terms of physical function, no differences were observed in any of the physical function measurements pre-intervention. The time taken to complete the 5X-STS significantly decreased in both the RET-PL (−8.1%, *p* < 0.001, *d* = 0.71) and RET-FO (−9.6%, *p* < 0.001, *d* = 1.00) groups. Similarly, the time for the TUG test was substantially reduced in both the RET-PL (−7.9%, *p* < 0.001, *d* = 1.11) and RET-FO (−11.3%, *p* < 0.001, *d* = 1.75) groups. The time taken to complete the 6MW test decreased significantly in both the RET-PL (−7.9%, *p* < 0.001, *d* = 0.86) and RET-FO (−11.1%, *p* < 0.001, *d* = 1.33) groups from pre- to post-intervention. The number of repetitions in the 30S-STS significantly increased in both the RET-PL (+11%, *p* < 0.001, *d* = 1.2) and RET-FO (+12.4%, *p* < 0.001, *d* = 1.5) groups.

Regarding BP measurements, there was a significant time effect for SBP and group × time interaction for DBP and MAP. No differences were noted for SBP, DBP, and MAP between the groups at baseline. The RET-FO group exhibited a substantial decrease in SBP (−4.4%, *p* = 0.041, *d* = 0.81), DBP (−4.4%, *p* = 0.001, *d* = 0.79), and MAP (−4.4%, *p* = 0.004, *d* = 0.87) post-intervention while no notable change was observed in the RET-PL group.

### 3.2. Biochemical Data

Table 2 describes the values of TG in all experimental groups before and after the 8 wk intervention. There was a significant group x time interaction for TG levels. No notable difference was observed in TG concentrations at baseline. TG levels were substantially decreased in the RET-FO group (−13.2%, *p* < 0.001, *d* = 0.69), while no significant change was detected in the RET-PL group (*p* = 0.179). Values of inflammatory biomarkers in all experimental groups are illustrated in Figure 1. There was a significant group x time interaction for TNF-α and IL-6. There were no differences in TNF-α and IL-6 between the experimental groups at baseline. TNF-α tended to decrease in the RET-PL group (−7.2%, *p* = 0.063, *d* = 0.36) from baseline and markedly decreased in the RET-FO group (−24.6%, *p* < 0.001, *d* = 1.28). Moreover, the RET-FO group presented significantly decreased IL-6 (−10.9%, *p* = 0.011, *d* = 0.32), while the RET-PL group showed no notable changes (*p* = 0.668). There was no notable difference in TNF-α and IL-6 between the groups post-intervention. Oxidative stress biomarkers (MDA and 8-OHdG) are presented in Figure 2. There was a significant group x time interaction for 8-OHdG. No differences were found in MDA and 8-OHdG between the groups at baseline. However, the RET-FO group exhibited a significant reduction in MDA (−12.8%, *p* = 0.013, *d* = 0.47) and 8-OHdG (−23.6%, *p* < 0.001, *d* = 0.5), with no detectable changes observed in the RET-PL group (*p* = 0.811, *p* = 0.238, respectively). There was no remarkable difference in MDA and 8-OHdG between the groups post-intervention.

## 4. Discussion

The primary aim of the present study was to evaluate the efficacy of *n*-3 PUFA administration through FO supplement in combination with 8 wks of RET on physical function and cardiometabolic health in postmenopausal women. The key findings of this study demonstrated that FO administration over an 8 wk period of programmed RET amplifies the beneficial effects of RET on some aspects of muscle strength (i.e., handgrip strength) and positively impacted risk factors associated with cardiometabolic disorders, such as BP, fat metabolism, and systemic inflammation and oxidative stress in postmenopausal women.

### 4.1. Physical Function

Our data indicated that the programed RET tended to improve handgrip strength (+2%, *p* = 0.060, *d* = 0.10) in postmenopausal women, and this improvement was significantly enhanced by FO supplementation (+5.7%, *p* < 0.001, *d* = 0.30). Similarly, our previous study reported a significant increase in handgrip strength in healthy older male and female adults after 12 wks of programed RET (+5.7%, *p* = 0.007, *d* = 0.21), which was augmented by FO administration (+9.4%, *p* < 0.001, *d* = 0.53). While handgrip strength failed to reach a statistical significance with 8 wks of RET intervention alone in postmenopausal women, FO administration supplemented the RET-induced benefits on grip strength. This finding suggests FO as a potential ergogenic aid for improving muscle strength in older female adults. In line with our results, Robinson and colleagues reported a linear relationship between FO consumption and grip strength improvement in older adults in their cross-sectional study [31]. Further, Smith and colleagues found great improvement in grip strength in healthy older males and females after 6 months of *n*-3 PUFA administration [27].

It is well established that chronic RET counteracts the loss of muscle strength and diminished physical function in older adults. We observed remarkable improvements in all aspects of physical function after 8 wks of RET, and FO appeared to enhance the RET-induced benefit of functional capacity in older females, as indicated by the Cohen’s *d* values. RET-FO exhibited greater improvements in all aspects of physical functions. In our previous study, we found that aging-induced physical function impairments were reversed by RET, and the RET-induced improvement was further amplified by FO administration. The positive correlation between *n*-3 PUFA administration and the enhancement of physical function in older adults has been well documented. Frison and colleagues observed that older adults with higher plasma levels of *n*-3 PUFA had lower gait speed (0.63 m/s) [32]. Similarly, another study conducted by Abbatecola and colleagues demonstrated that elevated systemic *n*-3 PUFA concentrations tended to mitigate decreased physical function, while a higher *n*-6/*n*-3 PUFA ratio appeared to increase the risk of poor physical function and slower gait speed [33]. Additionally, Reinders et al. demonstrated that higher *n*-3 PUFA plasma levels protected older women from significantly reduced gait speed [34], suggesting a cross-sectional association between high *n*-3 PUFA levels and muscle size and strength [35].

### 4.2. Blood Pressure

We observed a significant decrease in all aspects of BP in the RET-FO group, while no detectable change was found in the RET-PL group. Despite the wealth of evidence highlighting aerobic exercise as an effective approach for managing hypertension, the impacts of RET on improving BP lack consistency. Previous research has demonstrated that chronic programmed RET (e.g., 23 to 24 weeks) can improve BP in older adults [36,37]. However, other studies have presented conflicting results. Wood and colleagues demonstrated that chronic RET did not impact BP in normotensive elderly participants [38]. Their findings align with Anton et al. who also found no BP effects after a 13 wk RET regimen in normotensive older individuals [39]. The intervention period of our study was 8 weeks, and we did not find any improvements in BP from the 8 wk RET intervention. Given this, it could be conjectured that relatively short-term RET intervention alone may not suffice to bring about BP improvements in normotensive older adults.

Conversely, a noteworthy reduction in BP was evident when RET was coupled with FO supplementation among older women. The escalation of BP with aging primarily results from a combination of heightened vascular stiffness within the major arteries and atherosclerotic alterations in the vessel wall, along with impaired endothelial function which is responsible for vascular homeostasis and BP regulation [40]. The impaired vasomotor function with aging could be improved by *n*-3 PUFA, leading to improved BP in older individuals. The enhancement of endothelial function with *n*-3 PUFA encompasses several pathways, including an increase in the synthesis of vasodilator-like nitric oxide (NO), heightened vascular responsiveness to these vasodilators, and a decrease in the molecular concentration of substances within blood vessels that contribute to increased arterial stiffness (e.g., thromboxane A2, cyclic endoperoxides, etc.) [41]. Incorporating *n*-3 PUFA into the diet or engaging in long-term supplementation often leads to enhanced NO production, improved endothelial function, and heightened vasodilation. For instance, dietary EPA appeared to enhance endothelium-dependent relaxation in the porcine coronary artery [42], and prolonged exposure to cultured porcine endothelial cells to EPA resulted in increased NO release [43]. In animal models, long-term *n*-3 PUFA administration restored diminished NO production in orchiectomized rat [44] and in rat models of Angil-induced hypertension [45]. Collectively, it could be speculated that the observed enhancements in BP by FO supplementation stem from improved arterial compliance accompanied by enhanced endothelial function among older adults. Further research is required to substantiate this speculation.

### 4.3. Biochemical Biomarkers Associated with Cardiometabolic Health

Our data indicated that no detectable change was induced by sole RET intervention in TG, inflammation (TNF-α and IL-6), and oxidative stress (MDA and 8-OHdG) levels. Although the positive effects of aerobic exercise on fat oxidation are well established, findings from previous research determining the effects of RET on fat metabolism are relatively inconsistent. Ormsbee and colleagues found a substantial increase in fat oxidation after RET in both lean and obese sedentary adults [46]. In addition, Tsuzuku et al. reported significantly decreased TG levels in healthy older adults in response to 12 wks of RET; this was consistent with findings from Fahlman et al. who also observed a remarkable decrease in TG levels in older females after 10 wks of RET [47]. However, Ng and colleagues demonstrated that 8 wks of RET did produce notable improvements in TG levels in older adults with diabetes mellitus [48]. Similarly, Phillips et al. found no notable changes in TG levels in young, middle-aged, and older individuals after 20 wks of progressive RET [49].

On the other hand, we discovered that daily *n*-3 PUFA administration during 8 wks of RET appeared to decrease TG levels in older female adults. The lipolytic property of *n*-3 PUFA supplements has been reported in previous studies. Green and colleagues demonstrated that 8 wks of *n*-3 PUFA supplementation (4 g/day) resulted in a significant reduction in plasma TG levels in healthy adults compared to the control group [50]. In addition, Kuszewski et al. demonstrated that 16 wks of FO supplementation substantially decreased serum TG levels in older sedentary overweight/obese adults [51]. These findings support an anti-lipogenic capacity of FO that may hold promise as a potential anti-obesity agent. Taken together, it could be inferred that combining *n*-3 PUFA administration with RET may provide older females a substantial stimulus to avert fat gain, ultimately safeguarding them from obesity and obesity-related comorbidities.

Women often experience a decline in their primary sex hormone, estrogen, which plays a role in enhancing lipid profiles. However, during menopause, this hormone gradually decreases, resulting in the loss of muscle mass and bone density [52]. Moreover, the reduction in the female sex hormone may be linked to elevated levels of inflammatory mediators such as TNF-α and IL-6, hastening aging-induced muscle wasting [53]. In our study, 8 wks of RET did not yield significant improvements in systemic inflammation. Although there was a tendency for TNF-α to improve (−7.2%, *p* = 0.063, *d* = 0.36), it did not reach statistical significance. This aligns with a previous study that demonstrated a 12 wk RET protocol had no significant effect on systemic IL-6 and TNF-α levels in older women [18].

On the other hand, when combined with RET, FO administration produced substantial improvements in pro-inflammatory mediators such as TNF- α and IL-6. The correlation between chronic low-grade inflammation and cardiovascular disease [54] and metabolic disorders [55] has been extensively elucidated. Elevated inflammation plays a crucial role in the development of various cardiovascular disorders such as myocardial infarction, hypertension, ischemic heart disorder, or coronary artery disease [56,57]. Risks of cardiometabolic disorders significantly increase with age. Postmenopausal women, in particular, are more susceptible to one or more comorbidities. Therefore, it is imperative to provide effective countermeasure regimens for them. Given this, the enhancement of inflammation mediated by FO supplementation can protect them from age-induced cardiometabolic disorders. As mentioned earlier, we also observed remarkable improvements in other cardiometabolic risk factors (i.e., SBP and TG) with the combination of RET and FO, further supporting the clinical efficacy of FO in improving cardiometabolic health in older females.

The connection between oxidative stress biomarkers and cardiometabolic disorders has been extensively documented in previous research. A meta-analysis demonstrated that individuals with cardiovascular diseases exhibited higher 8-OHdG concentrations than healthy cohorts [58]. Similarly, a study indicated that elevated levels of MDA could independently predict cardiovascular disease and mortality in heart failure patients [59,60]. Besides cardiovascular diseases, elevated levels of these oxidative stress mediators may also exacerbate metabolic disorders. Although the intricate mechanisms linking oxidative stress and metabolic disorders are yet to be fully understood, it is plausible to speculate that oxidative stress-induced complications impact the insulin signaling pathway. Studies have demonstrated that heightened oxidative stress contributes to impaired insulin resistance [61,62]. For instance, Kim and colleagues demonstrated a link between increased phthalates and insulin resistance through the elevation of MDA in older adults [63]. Furthermore, other studies have reported that escalated DNA damage facilitated by the up-regulation of 8-OHdG may contribute to the development of gestational diabetes by insulin resistance [64,65].

The effects of FO administration on improving cardiometabolic diseases have been comprehensively elucidated in previous studies. DHA enriched with FO has been shown to mitigate lipid peroxidation and improve insulin sensitivity in skeletal muscle [66]. Ghorbanihaghjo et al. [67] found a notable reduction in 8-OHdG levels in healthy cigarette smokers after 3 month FO supplementation. Likewise, Buonocore and colleagues demonstrated that *n*-3 PUFA administration for 8 weeks significantly reduced MDA and TNF-α levels in both trained athletes and sedentary adults [68]. In alignment with these findings, the present study also discovered significant improvements in systemic oxidative stress biomarkers (MDA and 8-OHdG) in mice administered with FO. Collectively, it could be speculated that the antioxidant property of FO supplements may offer protective benefits to older females, potentially preventing the development of cardiometabolic disorders. Future research is necessary to validate this speculation.

While the present study yielded valuable insights, it is essential to acknowledge its limitations. First, the participants were allowed to maintain their regular daily activity levels and dietary habits, which may produce some variations in the results. In addition, the placebo used for the present study may also exert effects on lipid metabolism and thus reduce TG levels in those with high TG. Additionally, we did not assess the blood omega-3 levels and body composition of the participants, which could provide more robust support for the speculations stemming from the results.

## 5. Conclusions

The present study illustrates that FO administration over an 8 wk period of programmed RET amplifies the beneficial effects of RET on muscle strength. Simultaneously, it helps mitigate risk factors linked to cardiometabolic disorders in older females. Particularly, the combination of FO administration with RET exhibits potent anti-inflammatory and antioxidant properties, providing potential protection against the onset of various geriatric syndromes linked to chronic low-grade inflammation and/or aberrant oxidative stress in older women. The pivotal findings of the present study carry meaningful implications, serving as a foundation for forthcoming clinical investigations. Further studies are warranted to ascertain the clinical efficacy of FO treatment in counteracting the risk factors associated with sarcopenia and cardiometabolic disorders in postmenopausal women.

## Figures and Tables

**Figure 1 nutrients-15-04516-f001:**
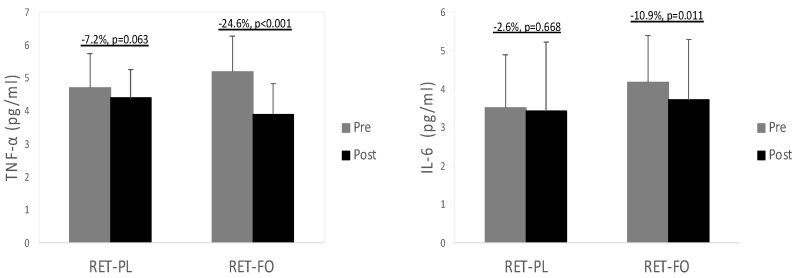
Systemic inflammation pre- and post-intervention. TNF-α = Tumor necrosis factor alpha; IL-6 = Interleukin-6. RET-PL = resistance exercise training with placebo; RET-FO = resistance exercise training + fish oil. Values are mean ± SD.

**Figure 2 nutrients-15-04516-f002:**
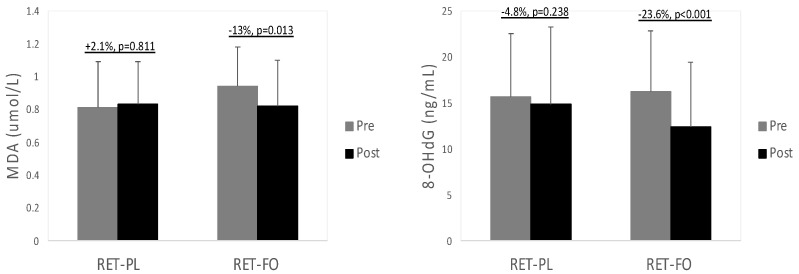
Blood oxidative stress biomarkers pre- and post-intervention. MDA = Malondialdehyde; 8-oHdG = 8-hydroxy-2′-deoxyguanosine. RET-PL = resistance exercise training with placebo; RET-FO = resistance exercise training + fish oil. Values are mean ± SD.

**Table 1 nutrients-15-04516-t001:** Descriptive characteristics of subjects at baseline. RET-PL = resistance training with placebo; RET-FO = resistance training + fish oil. Values are mean ± SD.

	RET-PL	RET-FO
Age (years)	65.4 ± 2.3	65.9 ± 4.3
Height (cm)	162.4 ± 4.9	166.7 ± 5.3
Weight (kg)	64.2 ± 5.6	66.3 ± 4.2
Body mass index (kg/m^2^)	24.4 ± 3.0	23.9 ± 1.5

**Table 2 nutrients-15-04516-t002:** Physical function, blood pressure, and triglyceride pre- and post-intervention. 5X-STS = five times sit-to-stand; TUG = timed up and go; 6MW = 6 m walk; 30S-STS = 30 s sit-to stand; SBP = systolic blood pressure; DBP = diastolic blood pressure; MAP = mean arterial pressure; TG = triglyceride. RET-PL = resistance exercise training with placebo; RET-FO = resistance exercise training + fish oil. Values are mean ± SD.

	RET-PL		RET-FO	
	Pre	Post	Pre	Post
Handgrip strength (kg)	20.4 ± 4.0	20.8 ± 4.3	20.9 ± 3.9	22.1 ± 4.0 *
5X-STS (second)	7.4 ± 0.9	6.8 ± 0.8 *	7.3 ± 0.7	6.6 ± 0.7 *
TUG (second)	6.3 ± 0.4	5.8 ± 0.5 *	6.2 ± 0.5	5.5 ± 0.3 *
6MW (second)	3.8 ± 0.3	3.5 ± 0.4 *	3.6 ± 0.3	3.2 ± 0.3 *
30S-STS (repetition)	19.1 ± 1.7	21.2 ± 1.8 *	19.4 ± 1.8	21.8 ± 1.4 *
SBP (mm Hg)	124.9 ± 4.9	123.7 ± 6.8	123.8 ± 5.1	118.3 ± 8.4 *
DBP (mm Hg)	81.5 ± 2.6	82.2 ± 3.2	81.0 ± 4.6	77.5 ± 4.3 *
MAP (mm Hg)	95.6 ± 3.2	96.0 ± 3.8	95.3 ± 4.6	91.1 ± 5.1 *
TG (mg/dL)	104.7 ± 12.6	101.7 ± 12	114 ± 22.4	99 ± 21.4 *

* *p* ≤ 0.05, significantly different from pre-intervention.

## Data Availability

The data presented in this study are available on request from the corresponding author. The data are not publicly available due to privacy.

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
