# Peer review of "Fish Oil Supplementation with Resistance Exercise Training Enhances Physical Function and Cardiometabolic Health in Postmenopausal Women"

_nutrients, 2023, doi:10.3390/nu15214516_

Round 1
Reviewer 1 Report
The manuscript submitted for review concerns an important topic.
However, the following issues should be clarified and clarified:
- qualification criteria: people have never taken omega-3 acids? Or only during a specific period?
- on what basis was the composition and proportion of EPA:DHA selected?
- was it a commercially available dietary supplement?
- why was an unusual oil used as a placebo: safflower?
- on which arm was blood pressure measured? On both to assess the differences?
- why did the authors select a group containing only 20 people? On what basis was the group size estimated?
Author Response
Dear Journal Editor and Reviewer,
We sincerely appreciate your efforts for another review. Please find our responses to your comments below.
Reviewer Comments #1
- Qualification criteria: people have never taken omega-3 acids? Or only during a specific period?
- Answer – We added the sentences as follow: ‘had not taken n-3 PUFA supplements in the 12 months that preceded in the present study (in lines 85-86).
- On what basis was the composition and proportion of EPA:DHA selected?
- Answer – The composition and proportion of EPA:DHA selected was stated and was approved by the FDA (in lines 126-128).
- Was it a commercially available dietary supplement?
- Answer – It is a commercially available dietary supplement. However, we made an agreement with the supplement company not to reveal their product information.
- Why was an unusual oil used as a placebo: safflower?
- Answer – Previous studies used safflower supplements as a placebo when testing the efficacy of fish oil supplements on the primary outcomes of the present study (e.g., inflammation, oxidative stress, blood pressure, etc.), which is aligned with the aims of the present study.
- On which arm was blood pressure measured? On both to assess the differences?
- Answer – The blood pressure was measured from the participants left arm and was done in a similar manner for all participants. We have added the sentences as follow: ‘Each participant sat in a quiet environment for 10 minutes, and the BP of their left arm, positioned on a table at heart level, was measured twice’ (in lines 160-161).
- Why did the authors select a group containing only 20 people? On what basis was the group size estimated?
- Answer – We added sentences as follows: ‘Group sizes were determined by a-priori power analysis using effect size calculated from previous evidence which showed notable differences in muscle strength following exercise in older adults [28]’ (in lines 80-82).

Reviewer 2 Report
The manuscript provided has potential, but requires extensive edits. Please carefully assess the methods, providing additional detail to ensure the scientific process was upheld, and avoid all conjecture in the discussion surrounding outcomes not measured in this investigation.
Introduction
While it is understandable that the authors may discuss the various physiological aspects of aging provided the subject pool utilized in the paper, this leads to unnecessary background unrelated to the specific aim of the paper, making it difficult to determine the purpose of the paper until the final sentence. The introduction should be restructured to remove mention of outcomes not assessed in this investigation. For example, lipid metabolism/profiles and dyslipidemia are not assessed. Discussion around these biomarkers should be removed, instead only focusing on the role of triglycerides as this was assessed. I would urge the authors to consider organizing the introduction to address each health concern (sarcopenia, inflammation, cardiovascular disease) individually and then explain their relationships to each other and how they are modulated by estrogen/aging.
Methods
Please explain how long subjects had not been engaging in resistance training (ex: 1 year vs 1 month), if any subjects were prescribed medications, what physical and mental disorders were considered as exclusionary vs not, and if subjects were screened for any other supplement use outside of n-3 PUFAs, such as antioxidants. Was diet, hydration, or sleep accounted for prior to examination days? Were biochemical assessments completed while fasted and for how long if so? If not directly monitored, were subjects instructed in the protocol to follow any specific guidelines regarding these factors outside of what is stated in lines 94-98? What food or beverages were they told would affect their metabolic rate? This direction seems unclear and prone to interpretation by the participant.
Information regarding the supplement requires more detail. Name of the manufacturer and any other nutrients included in the supplement is required at minimum.
The mention that the dose of EPA/DHA has received approval from the FDA to reduce triglyceride concentrations is not at all mentioned in the accompanied reference. This reference needs to be replaced with one that specifically demonstrated FDA support of this dose, or this sentence needs to be removed.
Results
Provide the actual p-values for the RET groups rather than simple stating they are >0.05.
Line 325, no effect to what markers exactly?
Discussion
All mention of changes to fat metabolism must be removed unless you are missing an entire methods section explaining that you conducted indirect calorimetry on subjects to assess fat oxidation or utilized a similar method, in which case, this must be added. You did not measure this! All paragraphs discussing fat oxidation (lines 337-359) must be removed as this is complete conjecture.
The discussion only highlights literature agreeing with no change to inflammatory markers from RET, but there is a wide body of evidence suggesting otherwise, this should be addressed in the discussion.
Author Response
Dear Journal Editor and Reviewer,
We sincerely appreciate your efforts for another review. Please find our responses to your comments below.
Reviewer Comments #2
- Introduction While it is understandable that the authors may discuss the various physiological aspects of aging provided the subject pool utilized in the paper, this leads to unnecessary background unrelated to the specific aim of the paper, making it difficult to determine the purpose of the paper until the final sentence. The introduction should be restructured to remove mention of outcomes not assessed in this investigation. For example, lipid metabolism/profiles and dyslipidemia are not assessed. Discussion around these biomarkers should be removed, instead only focusing on the role of triglycerides as this was assessed. I would urge the authors to consider organizing the introduction to address each health concern (sarcopenia, inflammation, cardiovascular disease) individually and then explain their relationships to each other and how they are modulated by estrogen/aging.
- Answer – Thank you for your comment. We revised the sentences according to the reviewer’s comments in the introduction (in lines 36-38). It reads as follows: ‘Research demonstrates a negative correlation between estrogen levels and triglyceride (TG) concentrations, with a notable increase in TG levels during menopause [4]’. Also, we removed the sentences that address the potential mechanisms underlying fat metabolism while the discussion regarding TG was left in the paper.
- Methods Please explain how long subjects had not been engaging in resistance training (ex: 1 year vs 1 month), if any subjects were prescribed medications, what physical and mental disorders were considered as exclusionary vs not, and if subjects were screened for any other supplement use outside of n-3 PUFAs, such as antioxidants. Was diet, hydration, or sleep accounted for prior to examination days? Were biochemical assessments completed while fasted and for how long if so? If not directly monitored, were subjects instructed in the protocol to follow any specific guidelines regarding these factors outside of what is stated in lines 94-98? What food or beverages were they told would affect their metabolic rate? This direction seems unclear and prone to interpretation by the participant.
- Answer – While we did not address the detailed guidelines for participants in the original draft, we provided a specific guideline for participants prior to each lab visits. We added the sentences according to the reviewer’s comment as follows. ‘Throughout the 8-wk intervention period, all participants were instructed to maintain their regular diet, normal daily activity, and regular sleep schedule’ (in lines 98-100). ‘They were advised to refrain from any strenuous exercise or physical activity (e.g., resistance exercise training held outside of the research laboratory) and to avoid consuming food (e.g., high-fat diet) or beverages (e.g., coffee, tea, etc.) that could affect their metabolic rate for 48 hours before their lab visit to minimize thevariability of the dependent variables’ (in lines 100-104). ‘Participants were instructed to fast at least eight hours prior to their scheduled blood draw for the biochemical assessments’ (in lines 169-170).
- Information regarding the supplement requires more detail. Name of the manufacturer and any other nutrients included in the supplement is required at minimum.
- Answer- While it is commercially available dietary supplement, we made an agreement with the supplement company not to reveal their product information.
- The mention that the dose of EPA/DHA has received approval from the FDA to reduce triglyceride concentrations is not at all mentioned in the accompanied reference. This reference needs to be replaced with one that specifically demonstrated FDA support of this dose, or this sentence needs to be removed.
- Answer – We apologize for the error. We changed the reference with one that demonstrated FDA support of the dose of FO that effectively improved TG.
- Results Provide the actual p-values for the RET groups rather than simple stating they are >0.05.
- Answer – We provided the actual p-values for the RET groups on the figures.
- Line 325, no effect to what markers exactly?
- Answer – We added the sentences to address the review’s comments as follow in lines 344-345, ‘Our data indicated no detectable change was induced by sole RET intervention in TG levels, inflammation (TNF-α and IL-6), and oxidative stress (MDA and 8-OHdG).
- Discussion All mention of changes to fat metabolism must be removed unless you are missing an entire methods section explaining that you conducted indirect calorimetry on subjects to assess fat oxidation or utilized a similar method, in which case, this must be added. You did not measure this! All paragraphs discussing fat oxidation (lines 337-359) must be removed as this is complete conjecture.
Answer – As responded to the first comment, we removed the sentences that address the potential mechanisms underlying fat metabolism. We kept the sentences in lines 407-417 that addressed n-3 PUFA administration appeared to decrease TG levels in older female adults with previous evidence that support our findings.

Reviewer 3 Report
It aims to determine the efficacy of n-3PUFA administration during 8-wks of programmed resistance exercise training on physical function, blood pressure, fat metabolism, and systemic inflammation and oxidative stress biomarkers in postmenopausal women. Its hypothesis was that n-3 PUFA administration would enhance the benefits derived from 8-wks of programmed resistance exercise training in older females.
There are a lot of studies on the effects of omega 3 fatty acid dietary supplementation on physical function, fat metabolism, systemic inflammation and oxidative stress in adults, males and females and in sportsmen and in sportswomen that not mentioned in the introduction and discussion; however, this information is limited in postmenopausal women. To focus the study in this population is very interesting and adequate as objective.
The experimental design is adequate to attain the objectives, but the use of placebo capsules is difficult to discover in the redaction of the manuscript. There is a lack of information on omega 3 intake in the two study groups. Which the dietary intake of omega-3 was in these groups? Normal dietary intake of omega-3 fatty acids is unknown, so the intake of omega-3 fatty acids by capsules might not be relevant compared to normal dietary intake; additionally, dietary intakes of omega-3 fatty acids between the two study groups may be different. You could include this information in the results section.
Minor questions
Lines 91-92: ‘participants were randomly assigned to either resistance exercise training alone (RET, n=10) or RET combined with FO supplementation (RET-FO, n=10) for a 8-wk intervention’ It seems that RET group only training for 8 weeks; NO placebo was intake. It was not a double blind study; but in the supplement section indicates the existence of a safflower oil capsules as placebo. Which the safflower oil fatty acid composition was? Clarify the use of placebo capsules and indicate the fatty acid and antioxidant composition of placebo and FO capsules.
Lines 96-98. Which the strenuous exercise or physical activity and foods or beverages could affect their metabolic rate for 48 hours?
Indicate the ethical committee approving this trial protocol and a reference code of this approbation.
Line 145: Are the words ‘6-minute line’ correct?
The physical function assessment evaluates the same functions using different test. Could you indicate the functions measured (strength, flexibility, resistance, speed, etc.) with each test?
Lines 174-182: The Cohen analysis can’t evaluate the effect size between supplemented and not-supplemented group. The effect of FO supplementation has to evaluate using the two-way ANOVA test as you analyse.
Table 1: The resistance training group also intake placebo capsules; perhaps will more adequate introduce this condition in the name of the group (RET-PL) as you do in the name RET-FO to the resistance training + fish oil group.
Lines 197-214 and Table 2: As you indicate in the statistics a two-way ANOVA was employed. Can you indicate the existence of the significance of the time or of the fish oil supplementation or interaction between these factors in the table 2 and in the text. It seems that RET induces significant changes both in RET and RET_FO, but no differences between RET and RET_FO values are present. It could be relevant the significance of the interaction between these factors to conclude about the effects of FO supplementation on the physical function of training menopausal women.
Indicate the results of the Two-way ANOVA analysis in the figures 1 and 2
Lines 250-254. The sentence 250-254 has to be concordant with the experimental design used in this study. The results did not demonstrate that FO administration enhances some aspects of muscle strength (i.e., handgrip strength) and positively impacted risk factors associated with cardiometabolic disorders, such as BP, fat metabolism, and systemic inflammation and oxidative stress in postmenopausal women. The study includes the resistance training in the experimental design, the effects observed can’t attributed only to the FO diet supplementation. The concept of ‘amplifies’ put in the conclusion (FO administration over an 8-wk period of programmed RET amplifies the beneficial effects of RET on muscle strength…) is more adequate than the sentence 250-254.
The discussion of results obtained supports the conclusions, but the additional study limitation has to be considered. Given the dose of safflower oil used as placebo was choice because effectively reduces TG concentrations in individuals with high TG levels, it will be adequate discuss the results obtained in the RET group. The fatty acid content of placebo can exert effects on lipid metabolism. Can you discuss it in your manuscript? Additional study limitations have to be considered referred to the knowledge of the total dietary intake of omega3 by the participants of both groups during the study and to the knowledge of the omega 3 content in the placebo.
Author Response
Dear Journal Editor and Reviewer,
We sincerely appreciate your efforts for another review. Please find our responses to your comments below.
Reviewer Comments #3
- The experimental design is adequate to attain the objectives, but the use of placebo capsules is difficult to discover in the redaction of the manuscript. There is a lack of information on omega 3 intake in the two study groups. Which the dietary intake of omega-3 was in these groups? Normal dietary intake of omega-3 fatty acids is unknown, so the intake of omega-3 fatty acids by capsules might not be relevant compared to normal dietary intake; additionally, dietary intakes of omega-3 fatty acids between the two study groups may be different. You could include this information in the results section.
- Answer – Thank you for your comments. We described composition of omega-3 in the RET-FO group in 2.3. fish oil supplement under methods. While each participant was instructed to consume their normal diet, the dose of omega-3 fatty acids used in the present study is approved by FDA to effectively reduce triglyceride levels (in lines 126-128). We also acknowledged that maintaining their regular diet which may produce some variation in the results, and we did not assess blood omega-3 levels, which could provide more robust support for the speculations stemming from the results (in lines 423 – 425).
- Lines 91-92: ‘participants were randomly assigned to either resistance exercise training alone (RET, n=10) or RET combined with FO supplementation (RET-FO, n=10) for a 8-wk intervention’ It seems that RET group only training for 8 weeks; NO placebo was intake. It was not a double blind study; but in the supplement section indicates the existence of a safflower oil capsules as placebo. Which the safflower oil fatty acid composition was? Clarify the use of placebo capsules and indicate the fatty acid and antioxidant composition of placebo and FO capsules.
- Answer – Lines 91-92 (now lines 94-97) were edited to clarify that the RET group was resistance exercise training and a placebo supplement and not resistance exercise training alone. Previous studies have used the safflower oil supplements as a placebo when testing the efficacy of fish oil supplements on the primary outcomes of the present study (e.g., inflammation, oxidative stress, blood pressure, etc.), which is aligned with the present study’s aims.
- Lines 96-98. Which the strenuous exercise or physical activity and foods or beverages could affect their metabolic rate for 48 hours?
- Answer – Strenuous exercise and physical activity were edited to be defined as resistance exercise training performed outside of the research laboratory that is outside of the research study. The section was also edited to address that the food that may alter metabolic rates are starting new diets from their regular dietary intake (high-fat diets etc), while beverages that may alter metabolic rates are drinks such as coffee and tea (in lines 100-104).
- Indicate the ethical committee approving this trial protocol and a reference code of this approbation.
- Answer – The IRB approval code and date were edited in lines 78-79 and the “Institutional Review Board Statement Section”, and “Informed Consent Statement” section (in lines 445-450).
- Line 145: Are the words ‘6-minute line’ correct?
- Answer – We apologize for the typo. The word was corrected to 6-meters.
- The physical function assessment evaluates the same functions using different test. Could you indicate the functions measured (strength, flexibility, resistance, speed, etc.) with each test?
- Answer – We changed the sentences as follow (in lines 143-144). ‘Physical function related to leg strength and activities of daily living was evaluated using the following tests’.
- Lines 174-182: The Cohen analysis can’t evaluate the effect size between supplemented and not-supplemented group. The effect of FO supplementation has to evaluate using the two-way ANOVA test as you analyse.
- Answer – Thank you for your comment. Based on our understanding, Cohen’s d is a suitable metric for assessing effect size when two groups exhibit have equal sample size and similar standard deviations. It has been recommended to accompany reporting of t-test and ANOVA results.
- Table 1: The resistance training group also intake placebo capsules; perhaps will more adequate introduce this condition in the name of the group (RET-PL) as you do in the name RET-FO to the resistance training + fish oil group.
- Answer – We revised the name of the RET to RET-PL throughout the entire manuscript.
- Lines 197-214 and Table 2: As you indicate in the statistics a two-way ANOVA was employed. Can you indicate the existence of the significance of the time or of the fish oil supplementation or interaction between these factors in the table 2 and in the text. It seems that RET induces significant changes both in RET and RET_FO, but no differences between RET and RET_FO values are present. It could be relevant the significance of the interaction between these factors to conclude about the effects of FO supplementation on the physical function of training menopausal women.
- Answer – We added the existence of the significance of factors and interaction between these factors in the sentences (in lines 247-248, 251-253, 277, 281-282, 278).
- Indicate the results of the Two-way ANOVA analysis in the figures 1 and 2
- Answer – We added the sentences for the result of the Two-way ANOVA for the figures in lines 245-246 and 251-252.
- Lines 250-254. The sentence 250-254 has to be concordant with the experimental design used in this study. The results did not demonstrate that FO administration enhances some aspects of muscle strength (i.e., handgrip strength) and positively impacted risk factors associated with cardiometabolic disorders, such as BP, fat metabolism, and systemic inflammation and oxidative stress in postmenopausal women. The study includes the resistance training in the experimental design, the effects observed can’t attributed only to the FO diet supplementation. The concept of ‘amplifies’ put in the conclusion (FO administration over an 8-wk period of programmed RET amplifies the beneficial effects of RET on muscle strength…) is more adequate than the sentence 250-254.
- Answer – Thank you for your comment. We changed the sentence as the reviewer suggested in lines 309-314. Now it reads as follow. The key findings of this study demonstrated that FO administration over an 8-wk period of programmed RET amplifies the beneficial effects of RET on some aspects of muscle strength (i.e., handgrip strength) and positively impacted risk factors associated with cardiometabolic disorders, such as BP, fat metabolism, and systemic inflammation and oxidative stress in postmenopausal women.
- The discussion of results obtained supports the conclusions, but the additional study limitation has to be considered. Given the dose of safflower oil used as placebo was choice because effectively reduces TG concentrations in individuals with high TG levels, it will be adequate discuss the results obtained in the RET group. The fatty acid content of placebo can exert effects on lipid metabolism. Can you discuss it in your manuscript? Additional study limitations have to be considered referred to the knowledge of the total dietary intake of omega3 by the participants of both groups during the study and to the knowledge of the omega 3 content in the placebo.
- Answer – Thank you for your comments. We added sentences as follows in lines 421-425, ‘Also, the placebo used for the present study may also exert effects on lipid metabolism and thus reduce TG levels in those with high TG. Additionally, we did not assess the blood omega-3 levels and body composition of the participants, which could provide more robust support for the speculations stemming from the results’.

Round 2
Reviewer 1 Report
The authors answered all my questions.
Reviewer 2 Report
Excellent work editing the manuscript! All concerns have been addressed.